# Association of Sarcopenia and Gut Microbiota Composition in Older Patients with Advanced Chronic Kidney Disease, Investigation of the Interactions with Uremic Toxins, Inflammation and Oxidative Stress

**DOI:** 10.3390/toxins13070472

**Published:** 2021-07-08

**Authors:** Elisabetta Margiotta, Lara Caldiroli, Maria Luisa Callegari, Francesco Miragoli, Francesca Zanoni, Silvia Armelloni, Vittoria Rizzo, Piergiorgio Messa, Simone Vettoretti

**Affiliations:** 1Division of Nephrology, Dialysis and Renal Transplantation, Fondazione IRCCS Cà Granda Ospedale Maggiore Policlinico, 20122 Milano, Italy; elisabetta.margiotta@policlinico.mi.it (E.M.); lara.caldiroli@policlinico.mi.it (L.C.); francescazanoni70@gmail.com (F.Z.); silvia.armelloni@policlinico.mi.it (S.A.); piergiorgio.messa@unimi.it (P.M.); 2Centro di Ricerche Biotecnologiche, Università Cattolica del Sacro Cuore, 26100 Cremona, Italy; marialuisa.callegari@unicatt.it (M.L.C.); francesco.miragoli@unicatt.it (F.M.); 3Laboratory Medicine and Clinical Biochemical Analysis, Ospedale San Matteo di Pavia, 27100 Pavia, Italy; v.rizzo@smatteo.pv.it; 4Department of Clinical Sciences and Community Health, University of Milan, 20122 Milan, Italy

**Keywords:** chronic kidney disease, sarcopenia, gut microbiota, inflammation, oxidative stress

## Abstract

Sarcopenia is a prevalent condition in chronic kidney disease (CKD). We determined gut microbiota (gMB) composition in CKD patients with or without sarcopenia. Furthermore, we investigated whether in these patients, there was any association between gMB, uremic toxins, inflammation and oxidative stress. We analyzed gMB composition, uremic toxins (indoxyl sulphate and p-cresyl sulphate), inflammatory cytokines (interleukin 10, tumor necrosis factor α, interleukin 6, interleukin 17, interleukin 12 p70, monocyte chemoattractant protein-1 and fetuin-A) and oxidative stress (malondialdehyde) of 64 elderly CKD patients (10 < eGFR < 45 mL/min/1.73 m^2^, not on dialysis) categorized as sarcopenic and not-sarcopenic. Sarcopenia was defined according to European Working Group on Sarcopenia in Older People 2 criteria. Sarcopenic patients had a greater abundance of the *Micrococcaceae* and *Verrucomicrobiaceae* families and of *Megasphaera*, *Rothia*, *Veillonella*, *Akkermansia* and *Coprobacillus* genera. They had a lower abundance of the *Gemellaceae* and *Veillonellaceae* families and of *Acidaminococcus* and *Gemella* genera. GMB was associated with uremic toxins, inflammatory cytokines and MDA. However, uremic toxins, inflammatory cytokines and MDA were not different in sarcopenic compared with not-sarcopenic individuals, except for interleukin 10, which was higher in not-sarcopenic patients. In older CKD patients, gMB was different in sarcopenic than in not-sarcopenic ones. Several bacterial families and genera were associated with uremic toxins and inflammatory cytokines, although none of these latter substantially different in sarcopenic versus not-sarcopenic patients.

## 1. Introduction

Sarcopenia is defined as a loss of skeletal muscle mass and function. In Chronic Kidney Disease (CKD) patients, sarcopenia is highly prevalent and it is associated with physical disability, low quality of life and increased mortality [1]. In these patients, sarcopenia is determined by a series of complex mechanisms, such as malnutrition, systemic inflammation, uremic toxins retention, metabolic acidosis, vitamin D deficiency, insulin resistance and some other hormonal factors often present in CKD patients [2].

In CKD patients, retention of uremic molecules and pro-inflammatory cytokines is the key mechanism leading to oxidative stress and inflammation [3]. In particular, two uremic toxins, indoxyl sulphate (IS) and p-cresyl sulphate (PCs), are produced by proteolytic bacterial fermentation in the gut [4,5] and retained in the serum of CKD patients owing to both increased intestinal production and reduced glomerular filtration and proximal tubular secretion [6,7,8]. IS has been shown to be associated with aortic calcification and vascular smooth muscle cell proliferation in rats and humans [6,9,10]; elevated PCs has been associated with insulin resistance [7] and vascular disease [8]. Both toxins play an important role in inducing muscular atrophy through various mechanisms (e.g., pro-inflammatory cytokines release or activation of some pathways, such as aryl hydrocarbon receptor (AhR), NFκB and SRAA) [11]. IS contributes to skeletal muscle loss by inhibiting myoblast proliferation through an enhanced expression of myostatin and atrogin-1 mediated by oxidative stress [11]. Moreover, IS promotes metabolic alterations with consequent mitochondrial dysfunction and suppression of anabolism signaling and ATP production [11]. Finally, IS induces atrophy of myotubes through the production of reactive oxygen species [12]. In patients with end stage renal disease in dialysis, there is a significant inverse association between plasma IS and skeletal muscle mass [11], but this association was not observed in older patients with advanced CKD not yet in dialysis [13].

Pathophysiological pathways by which PCs may influence the onset of sarcopenia are different. In a murine model, Koppe et al. demonstrated that PCs inhibits insulin-stimulated glucose uptake and, by activating ERK kinase, decreases insulin signaling pathways. Moreover, PCs suppresses insulin-induced phosphorylation of Akt, decreasing muscle protein synthesis and inducing protein degradation [7]. Therefore, it is likely that IS and PCs may induce the onset of sarcopenia by interfering with different metabolic pathways.

In addition to the role played by uremic toxins, it has recently been demonstrated that intestinal dysbiosis and gastrointestinal barrier disruption may significantly contribute to sustaining a low-grade systemic inflammation in CKD [11,12]. In CKD patients, gut dysbiosis is characterized by distinctive qualitative changes, such as increased the Enterobacteriaceae and reduced the *Lactobacillaceae* and *Prevotellaceae* families [13,14,15]. In our previous study, in which we compared CKD patients with healthy controls, we found that the former were characterized by increased abundance of the *Citrobacter*, *Anaerotruncus*, *Coprobacillus* genera and *Ruminococcus* torques species and reduced abundance of saccharolytic and butyrate-producing bacteria (*Prevotella*, *F. prausnitzii*, *Roseburia*) [16].

Furthermore, we observed several relevant differences in the gMB composition between frail and not-frail CKD patients, where frail patients showed higher abundance of the *Lactobacillus*, *Oscillospira*, *Eggerthella* (particularly *E. lenta* species), *Erwinia*, *Anaerotruncus*, *Coprobacillus* genera and the *Coriobacteriaceae* and *Mogibacteriaceae* families [16].

The principal aim of this study was to examine if in older patients with advanced CKD, the gMB composition may differ between sarcopenic and not-sarcopenic individuals. Secondly, we explored whether in these patients, there may be any association of gMB with uremic toxins, inflammatory cytokines and oxidative stress. Finally, we explored the association between these latter substances and sarcopenia in this population.

## 2. Results

### 2.1. Population Characteristics

We evaluated 64 patients with a median age of 80.7 ± 6.2 years, mostly males (69%). Thirty-seven patients were diabetic. They had a median eGFR of 26 ± 11 mL/min/1.73 m^2^ and a median BMI of 28.4 ± 4.7. Sarcopenia had a prevalence of 28% (18/64). Clinical parameters (age, presence of frailty, BMI, CKD-EPI GFR, MIS and PCR) did not differ between sarcopenic and not-sarcopenic patients, except for BMI, which was higher in the not-sarcopenic ones. (Table 1).

### 2.2. Sarcopenia and gMB Composition

We evaluated the gMB composition by examining a total of 8 phyla, 34 families and 52 genera. The statistically different families and genera between S and NS patients are shown in Figure 1 and Figure 2, while full analyses are reported in Appendix A. All data refer to direct correlations without any adjustment for other covariates.

In the sarcopenic group, there was a greater abundance of the *Micrococcaceae* (FDR = 0.012) and *Verrucomicrobiaceae* (FDR = 0.012) families and a lower abundance of the *Veillonellaceae* (FDR = 0.012) and *Gemellaceae* (FDR = 0.042) families (Figure 1). Regarding genera, sarcopenic subjects had greater abundance of *Megasphaera* (FDR < 0.001), *Veillonella* (FDR < 0.001), *Rothia* (FDR = 0.004) (Figure 2), *Coprobacillus* (FDR = 0.01) and *Akkermansia* (FDR = 0.008) and reduced abundance of *Acidaminococcus* (FDR < 0.0001) and *Gemella* (FDR = 0.03) (Figure 2).

#### 2.2.1. Correlations of gMB with Uremic Toxins

We found a direct correlation between IS and *Lactobacillus* genus (Table 2), unclassified *Lactobacillus* and *Bifidobacterium longum* species (Table 3) and the OTU *Bifidobacterium longum* (r = 0.28, *p* = 0.02) and *Bacteroides 183480* (r = 0,29; *p* = 0.024) (data not shown).

PCs was directly correlated with four genera—unclassified *Coriobacteriaceae*, *Desulfovibrio* (belonging to *Proteobacteriaceae* family), unclassified *Mogibacteriaceae* and *Christensenellaceae* (Table 2)—and two species—*Eubacterium biforme* and *Oscillospira* (Table 3).

#### 2.2.2. Correlations of gMB with Inflammatory Cytokines

Interleukin 10 was inversely associated with *unclassified Bacteroides* species (Table 4).

Interleukin 6 was directly associated with unclassified Bacteroides species (Table 4) and inversely correlated with *Bifidobacterium 825808* OTU (r = −0.25, *p =* 0.044). Interleukin 6 was directly associated only with Bacteroides species (Table 4).

Interleukin 17 was positively associated with the *Collinsella* genus (Table 3) and particularly with *Collinsella aerofaciens* species (Table 4) and the OTU *Collinsella aerofaciens 368175* (r = 0.3, *p* = 0.019).

Tumor necrosis factor alpha was directly associated with some genera—*unclassified Rikenellaceae*, *Bacteroides* and *unclassified Barnesiellacaeae*—and inversely correlated with *Streptococcus* and *Lactobacillus* genera (Table 3).

Interleukin 12 was inversely associated with *Eggerthella* and *Alistipes* genera (Table 3).

Monocyte chemoattractant protein-1 was negatively associated with *unclassified Barnesiellacaeae*, *Ruminococcus* (specifically with *Ruminococcus 523140* OTU, r= −0.28, *p* = 0.0232), *Bacteroides* and *Eggerthella* genera (Table 3).

Fetuin-A was directly associated with the *Collinsella* genus (Table 3) and *Collinsella aerofaciens* species (specifically with the OTU *Collinsella aerofaciens*_368175, r = 0.26, *p* = 0.0371; Table 3) and negatively associated with *Enterobacteriaceae*_797229 OTU (r = −0.28, *p* = 0.025).

#### 2.2.3. Correlations of gMB with Oxidative stress

We found a direct correlation between malondialdehyde and the abundance of three *Blautia*-related OTUs (*Blautia* 367790: r = 0.27, *p* = 0.031, *Blautia 570507*: r = 0.2, *p* = 0.03, *Blautia 316452*: r = 0.26, *p* = 0.038) and *Ruminococcaceae 361811* (r = 0.27, *p* = 0.03), and an inverse correlation between MDA and the abundance of the OTU *Bacteroides 183480* (r = −0.26, *p* = 0.036).

### 2.3. Sarcopenia, Uremic Toxins, Oxidative Stress and Inflammatory Parameters

Interleukin 10 concentration was higher in not-sarcopenic than in sarcopenic patients. There were no other differences in the concentrations of IS, PCs, MDA and all the analyzed cytokines and inflammatory parameters between sarcopenic and not-sarcopenic patients (Table 4).

## 3. Discussion

The aim of our study was to investigate the associations of the gMB composition with the presence of sarcopenia in older patients with advanced CKD. Secondly, we evaluated if there was any association between the gMB composition and uremic toxins, inflammatory or oxidative stress markers.

Overall twenty-eight% of our patients were sarcopenic. In this cohort BMI is significantly lower in sarcopenic individuals and this may depend on the correlation between sarcopenia and malnutrition that is quite a common characteristic in older patients with advanced CKD, as it was described also by our group [17].

Consistently with our previous results regarding sarcopenia in older CKD individuals, we did not find any correlation between BMI and inflammatory cytokines.

BMI was, instead, respectively directly and inversely correlated with IS and pCS. These results may at least partially depend on the correlations between poor nutritional status and sarcopenia in our patients. Indeed, in a previous study by our group conducted in older CKD patients, IS and pCS were oppositely associated with nutritional indices [13].

Regarding the gMB composition, we found a greater abundance of the *Micrococcaceae* and *Verrucomicrobiaceae* families in sarcopenic compared with not-sarcopenic patients. These families were previously found to be markedly increased in ESRD patients compared with healthy controls [15]. *Verrucomicrobiaceae* possess tryptophanase (an indole-producing enzyme, leading to increased IS production), while *Micrococcaceae* possess urease and uricase [18], which, through generation of high quantity of ammonia and ammonium hydroxide, play a key role in the development of uremic enterocolitis [19] and contribute to systemic inflammation [20]. *Verrucomicrobia* was more abundant in a mice model of chemotherapy-induced malnutrition and muscle-wasting [21]. Higher abundance of *Verrucomicrobia* was also associated with inflammatory diseases, such as primary sclerosing cholangitis [18,19], psoriasis and psoriatic arthritis [22], and with hepatic encephalopathy in cirrhotic patients, conditions often characterized by protein malnutrition and muscle wasting [23].

*Rothia* genus was more abundant in the sarcopenic group compared with the non-sarcopenic one. An overrepresentation of *Veillonella* and *Rothia* genera has been previously described on the skin of centenarians [24]. These genera have been associated not only with old age, but also with frailty. In frail residents of nursing homes, a shift towards dysbiosis was observed when compared with healthy active community dwellers. In particular, they may have an increased representation of pathobionts such as *Veillonella* in their salivary microbiota [25] and an increased relative abundance of *Rothia mucilaginosa* in their nasal microbiota [26]. The presence of these bacteria in salivary microbiota may influence the gMB composition and represent a risk factor for infections, especially pneumonia [27].

In our study, *Coprobacillus* and *Akkermansia* were more abundant in sarcopenic patients compared with not-sarcopenic ones.

*Coprobacillus* has been widely associated not only with biological aging and frailty [28,29] but also with inflammation [30] and hypertensive nephropathy [31].

*Akkermansia* is considered a beneficial bacterium due to its anti-inflammatory properties, and it has been found to be negatively correlated with inflammatory bowel disease [32], BMI and obesity [33,34], while it is positively associated with weight loss [35]. *Akkermansia* has been described in great abundance in animal models and in individuals with colorectal carcinomas and multiple sclerosis [28] and it is capable of exacerbating *Salmonella typhimurium*-induced intestinal inflammation by its ability to disturb host mucus homeostasis [29]. Moreover, some authors reported that the abundance of *Akkermansia* and other genera increased along with the progression of CKD, suggesting that this bacterium may play an important role in CKD progression [36]. In our study, *Akkermansia* was more abundant in sarcopenic patients, and it was inversely correlated with BMI (correlation with a low statistical strength: *p* = 0.03, r = –0.26). These results could possibly influence the inverse relationship linking BMI and sarcopenia [37,38]. However, we do not know if greater abundance of *Akkermansia* in sarcopenic individuals could also play a pro-inflammatory role influencing the onset of sarcopenia.

In our study, sarcopenic patients were also characterized by lower abundances of the *Veillonellaceae* and *Gemellaceae* families as well as of *Acidaminococcus* and *Gemella* genera when compared to not-sarcopenic patients. The *Gemellaceae* and *Veillonellaceae* families and *Gemella* genus have been associated with obesity [28,29]. *Acidaminococcus* species from the *Veillonellaceae* family may act as opportunistic pathogens [36,39] and can contribute to maintaining a healthy status through the production of short-chain fatty acids (SCFA), such as butyrate and acetate [40]. Moreover, decreased abundance of this genus has been described in CKD patients compared with healthy controls [14,33].

We observed a direct correlation between IS levels and tryptophanase-containing bacteria (*Lactobacillus* genus, *unclassified Lactobacillus* and *Bifidobacterium longum* species and the OTU *Bacteroides 183480*), which can process tryptophan into indole, which is then metabolized by the liver into IS [41,42]. Previously, we had already found that *Lactobacillus* abundance was higher in CKD patients compared with healthy controls [43].

We observed also positive correlations between PCs and bacteria that have been associated with several unhealthy conditions. Specifically, the *Coriobacteriaceae* family comprises many pathobionts involved in human infections, such as Bacteremia and periodontitis; moreover, *Coriobacteriaceae* family was significantly increased in the ceca of mice in response to stress [44]. In our previous study [43], the *Coriobacteriaceae* and *Mogibacteriaceae* families and the *Oscillospira* genus were more abundant in frail individuals and were positively correlated with markers of inflammation. *Oscillibacter*-like organisms (including *Oscillibacter* and *Oscillospira*) mediate high-fat diet-induced gut dysfunction [45], and *Oscillibacter* sp. are one of the few bacterial species capable of generating protein-bound uremic toxins, such as IS and PCs, from precursor metabolites under anaerobic conditions [41]. *Desulfovibrio* belongs to *Proteobacteriaceae* families that include a wide variety of pathogenic genera involved in diseases, such as cystic fibrosis [42]. *Christensenella* is generally considered a healthy bacterium [46], while in a large cohort of patients with early kidney disease, *Christensenellaceae* families were positively associated with IS and PCs levels [47].

Malondialdehyde (MDA) is an advanced oxidation product and a recognized oxidative stress biomarker whose levels are increased in patients with CKD [48]. In our cohort, we found a significant positive correlation between MDA and the presence of four OTUs of the genus *Blautia* and *Ruminococcaceae 361811* and an inverse correlation between MDA and the OTU *Bacteroides 183480*. An increase of *Blautia* has been observed in CKD patients [47]. Furthermore *Blautia* has been previously found to be associated with trimethylamine-N-oxide (TMAO) [49] and PCs [50]. Intestinal Bacteroidetes can have a double role in human health: they are known as a major source of propionate but are also involved in the release of toxic products from protein breakdown. Members of this group can exert activities that may help to suppress inflammation, but they also have the potential to promote inflammation and some of them are known to be opportunistic pathogens. In our results, we observed a negative correlation between the presence of OTU *Bacteroides 183480*, probably a beneficial strain, and MDA. In our study, an unclassified *Bacteroides* species, which belongs to the *Bacteroidetes* family, was negatively correlated with interleukin 10 and positively correlated with interleukin 6, anti-inflammatory and pro-inflammatory cytokines. Interleukin 6 was also negatively correlated with *Bifidobacterium 825808* OTU (known to have a probiotic role).

Interleukin 17 was positively correlated with the *Collinsella* genus and particularly with *Collinsella aerofaciens* species. The genus *Collinsella* is the dominant taxon of the family *Coriobacteriaceae*, commonly considered as pathobionts. Its abundance has been associated with type 2 diabetes [51], rheumatoid arthritis [52] and frailty [43].

We also found a direct correlation between tumor necrosis factor alpha, *Bacteroides, unclassified Rikenellaceae* and *unclassified Barnesiellaceae* species. Several species belonging to the *Rikenellaceae* family have been demonstrated to be pathogenic by promoting inflammation and producing mutagenic toxins [53]. *Barnesiella* is generally considered a healthy bacterium because it prevents pathogenic species of antibiotic-resistant bacteria from colonizing the gut.

Interleukin 12 is inversely correlated with the *Eggerthella* and *Alistipes* genera and with *Ruminococcus 523140* OTU. All these bacterial genera may have a useful role for human health: *Eggerthella lenta* is part of the normal human gMB, particularly abundant in frail and old subjects [15,50]. In our study, *Eggerthella* was found to be inversely associated also with monocyte chemoattractant protein-1. *Alistipes* may have protective effects against some diseases, including liver fibrosis, colitis and cardiovascular disease; however, its role is still debated [54]. *Ruminococcus*, among other bacteria, belongs to the *Clostridium* cluster IV, known for their ability to produce butyrate and traditionally considered as healthy bacteria, very important in maintaining intestinal homeostasis [55].

Monocyte chemoattractant protein-1 was found to be inversely associated with *unclassified Barnesiellacaeae, Ruminococcus* (specifically with *Ruminococcus 523140* OTU), *Bacteroides* and *Eggerthella* genera. In our previous study, *Eggerthella* was found to be associated with frailty [43] that may be reasonably considered a surrogate of sarcopenia.

Fetuin A was a directly associated with *Collinsella aerofaciens* species (and particularly with the OTU *Collinsella aerofaciens 368175*) and negatively correlated with *Enterobacteriaceae* 797229 OTU (Tab. 6). Fetuin A is considered as a negative acute phase reactant and its levels are inversely associated with serum CRP [56]. In our study, fetuin A was inversely associated with *Enterobacteriaceae 797229* OTU that are considered to be pro-inflammatory. Fetuin-A is also an inhibitor of insulin receptors and its higher levels have been previously associated with insulin resistance [54,57]. Insulin resistance is an early metabolic alteration in CKD patients and its prevalence is significantly increased in ESRD. Diabetic patients have higher abundance of *Collinsella* [58]. Therefore, it may be that the association between *Collinsella aerofaciens* and fetuin-A that we found in our patients may even influence insulin resistance.

According to our results, none of the correlations observed between gMB, uremic toxins and inflammatory markers seems to be associated with sarcopenia. We did not find any significant difference in blood levels of these markers between the sarcopenic and not-sarcopenic group, except for interleukin 10. However, although interleukin 10 is the only cytokine whose levels are statistically different between sarcopenic and not-sarcopenic individuals, its values do not correlate with any of the bacteria found to be associated with sarcopenia.

Our study has several limitations. Given its cross-sectional design, our results cannot be considered more than descriptive. However, our aim was to explore whether in older CKD patients, there was any difference in the gMB composition between sarcopenic and not-sarcopenic individuals and our results suggest that some differences do exist. The differences in the gMB composition that we found are only slightly significant, but this may depend on the fact that, in order to reduce the possible sources of bias, we applied stringent selection criteria; therefore, the clinical characteristics of our cohort were relatively homogeneous. Advanced age of our patients could make our results not completely representative of all patients with CKD, but we purposely focused our research on older individuals because they have the higher prevalence of sarcopenia.

Finally, the descriptive nature of the study prevents us from discerning if altered gMB is merely the consequence of the inflammatory and toxic milieu characterizing CKD, or if it can actually contribute to the development of sarcopenia in this population.

Overall, we believe that our results may help to formulate new hypotheses to investigate the association between sarcopenia and gut microbiota composition in older patients with advanced CKD.

## 4. Conclusions

Our results suggest that in elderly patients with advanced CKD, the gMB composition is significantly different between sarcopenic and not-sarcopenic individuals. Furthermore, in CKD patients, there are many mutual correlations between gMB composition, uremic toxins, inflammatory cytokines and oxidative stress parameters.

## 5. Materials and Methods

### 5.1. Population Characteristics

We cross-sectionally evaluated 64 patients affected by stage IIIb-IV CKD (10 < eGFR < 45 mL/min/1.73 m^2^, not on dialysis), aged ≥65 years, enrolled from those that were consecutively attending the outpatient clinic of the Unit of Nephrology at the Fondazione IRCCS Ca’ Granda Ospedale Maggiore Policlinico di Milano. Exclusion criteria were inflammatory diseases and/or ongoing immunosuppressive treatment, cancer, heart failure >NYHA II, decompensated liver disease, use of probiotics/antibiotics within 3 months before study enter and inability to collaborate.

The study was conducted according to the ICP Good Clinical Practices Guidelines and to the Declaration of Helsinki, and it was approved by the Ethics Committee of our Institution (approval number 347/2010, approved on 23 October 2010).

Recruitment of participants started on 1 September 2015 and ended on 6 December 2016. All eligible patients were screened during the observational period and were asked to participate in the study. All participants were requested to sign a written informed consent before being included in the study, as specified in the ICMJE recommendations.

We asked all enrolled patients to complete a questionnaire regarding antibiotics, probiotics and/or any immunosuppressive drugs used within 3 months before fecal sample collection. Each volunteer had to collect feces at home on the day preceding the visit, using 20 mL plastic sterile stool collection containers, and to place it in their own freezer at −18/−20 °C overnight, before it was stored at −80 °C in our laboratories until analysis.

Clinical evaluation and blood samplings were done in the morning after an overnight fasting of at least 10 h.

Sarcopenia was defined according to the criteria of the European Working Group on Sarcopenia in Older People 2 (EWGSOP2) [1].

### 5.2. Measurement of Uremic Toxins and Malondialdehyde

Uremic toxins and malondialdehyde dosing were performed at the laboratory of Biochemistry of the University of Pavia.

Free fraction of indoxyl sulphate and p-cresyl sulphate concentrations in serum were determined by high performance liquid chromatography (HPLC) and fluorescence detection (FLD) [59].

To determine malondialdehyde concentration in serum, we used the HPLC method with fluorescent detection based on the 2-thiobarbituric acid (TBA) assay, using the Chromsystems kit (Chromsystems Instruments & Chemicals GmbH, 82166, Gräfelfing, Germany) [60].

### 5.3. Measurement of Serum Cytokines

Enzyme-linked immunosorbent assay (ELISA) analyses have been assessed on serum samples at the Laboratory of nephrology of our Institution. Some specific kits were used: Human Interleukin 10 ELISA Kit EHIL10 (Invitrogen, Thermo Fisher Scientific, Monza, Italy), Quantikine ELISA Human CCL2/Monocyte chemoattractant protein-1 Immunoassay DCP00, Quantikine ELISA Human Interleukin 12 p70 Immunoassay D1200 (all R&D Systems, Space, Milano, Italy) Human tumor necrosis factor alpha ELISA Kit (Thermo Fisher Scientific, Monza, Italy). Quantikine ELISA Human Interleukin 17 Immunoassay, Quantikine HS ELISA Human Interleukin 6 Immunoassay HS600B (R&D Systems, Space, Milano, Italy).

### 5.4. Bacterial DNA Extraction and V3–V4 Region Sequencing

Bacterial DNAs were extracted from 50 mg of fecal sample using the FastDNA™ SPIN Kit for Soil (MP Biomedicals, Lucerna, Switzerland) according to the manufacturer’s instructions. The PCR amplifications were performed using the primers 343F and 802R. To allow the sequence demultiplex, a specific tag was attached to the forward primer. The PCR amplification and amplicon purification were performed as already described [16].

Sequencing was performed using Illumina’s MiSeq platform (Parco Tecnologico Padano, Lodi, Italy) with 300 bp paired-end mode and v3 chemistry and analyzed as previously described. [16]

### 5.5. Statistical Analysis

Results were expressed as mean ± SD or median ±IQR. Comparisons of normally distributed variables were done using Student’s *t*-test, while the comparisons of not normally distributed ones were done by using Mann–Whitney U test. Proportions and categorical variables were compared by using independent chi-squared (χ^2^) test or Fisher’s exact test. Regression analyses were performed by using Pearson or Spearman tests, as appropriate. Statistical analysis was carried out with the Statview software, version 5.0.1.

Sequence analysis were performed using MicrobiomeAnalyst tool [61,62] using the EdgeR algorithm with 0.05 false-discovery rate (FDR) cut-off for identifying significant differences in taxa abundance of bacteria between the two group of subjects.

Statistical significance was set at *p* < 0.05.

## Figures and Tables

**Figure 1 toxins-13-00472-f001:**
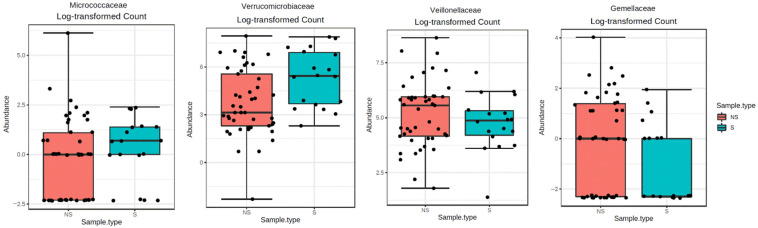
Comparison of statistically significant bacterial families between sarcopenic (S) and not sarcopenic (NS) patients.

**Figure 2 toxins-13-00472-f002:**
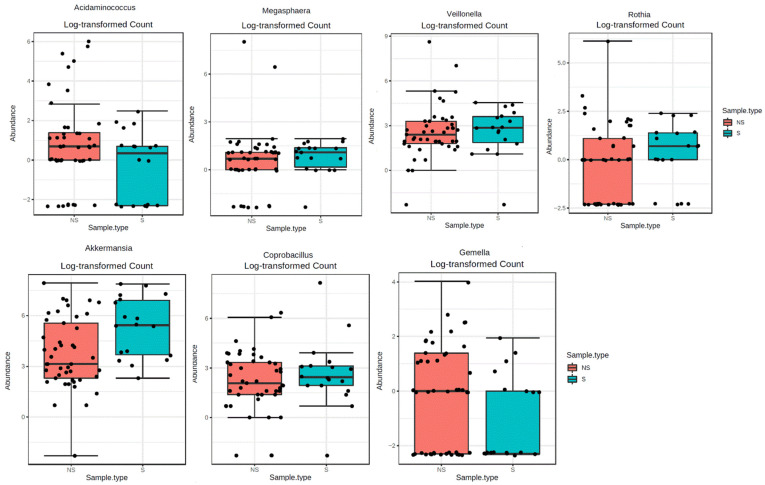
Comparison of statistically significant bacterial genera between sarcopenic (S) and not-sarcopenic (NS) patients.

**Table 1 toxins-13-00472-t001:** Clinical parameters in sarcopenic and not-sarcopenic individuals.

Variables	Overall Cohort (*n* = 63)	Sarcopenic (*n* = 18)	Not Sarcopenic (*n* = 45)	*p*
Age (years)	80.7 ± 6.2	83.1 ± 5.7	79.7 ± 6.2	0.0528
Males, *n* (%)	44 (69)	16 (89)	28 (62)	0.12
eGFR (ml/min/1,73 m^2^)	26 ± 11	23.9 ± 10	25.5 ± 10	0.5733
Diabetes, *n* (%)	37 (59%)	11 (61%)	26 (58%)	0.75
Frailty, *n* (%)	38 (59)	13 (72)	24 (53)	0.16
BMI (kg/m^2^)	28.4 ± 4.7	25.5 ± 2.6	29.3 ± 4.8	0.025
MIS	6 ± 4.7	6.7 ± 4.2	5.7 ± 4.9	0.48
CRP, (mg/dl)	0.3 ± 0.5	0.43 ± 0.9	0.28 ± 0.22	0.20
HCO_3_^–^	24.0 ± 3.5	24.0 ± 3.6	23.9 ± 2.9	0.93

eGFR, estimated glomerular filtration rate; BMI, Body Mass Index; MIS, Malnutrition Inflammation Score; CRP, C-reactive protein; HCO_3_^−^, serum bicarbonate.

**Table 2 toxins-13-00472-t002:** Association of gMB genera with uremic toxins, oxidative stress and inflammatory cytokines.

GENUS		Uremic Toxins		Anti-Inflammatory Cytokines			Pro-Inflammatory Cytokines				Oxidative Stress
		IS µmol/L	PCs µmol/L	IL-10 pg/mL	IL-12 pg/mL	Fetuin A ng/mL	IL-6 pg/mL	IL-17 pg/mL	TNF-α pg/mL	MCP-1 pg/mL	MDA µmol/L
*Alistipes*	r p	0.01 0.95	0.11 0.38	−0.05 0.70	**−0.25** **0.048**	−0.17 0.17	0.048 0.71	−0.23 0.07	0.077 0.54	−0.19 0.13	0.12 0.36
*Bacteroides*	r p	0.06 0.64	−0.1 0.44	−0.22 0.086	−0.04 0.76	−0.035 0.78	0.23 0.06		**0.27** **0.033**	**−0.29** **0.019**	0.11 0.40
*Collinsella*	r p	0.15 0.25	0.05 0.70	0.16 0.21	−0.039 0.76	**0.25** **0.045**	−0.09 0.47	**0.31** **0.013**	−0.14 0.27	0.023 0.86	0.021 0.87
*Desulfovibrio*	r p	−0.025 0.84	**0.29** **0.02**	−0.056 0.66	0.045 0.73	0.15 0.25	−0.11 0.39	−0.038 0.77	−0.09 0.498	0.095 0.45	0.046 0.72
*Eggerthella*	r p	0.007 0.95	−0.09 0.50	−0.13 0.29	**−0.26** **0.04**	−0.08 0.50	−0.07 0.69	−0.04 0.74	0.16 0.21	**−0.29** **0.022**	0.17 0.18
*Lactobacillus*	r p	**0.39** **0.002**	−0.07 0.58	0.17 0.18	0.049 0.71	0.11 0.40	−0.08 0.55	−0.05 0.72	**−0.34** **0.006**	0.08 0.51	−0.07 0.60
*Ruminococcus*	r p	−0.007 0.96	0.09 0.49	−0.12 0.34	−0.18 0.15	0.095 0.45	0.02 0.86	−0.15 0.25	0.08 0.53	**−0.33** **0.007**	0.17 0.19
*Streptococcus*	r p	0.18 0.16	0.06 0.64	0.17 0.17	−0.19 0.14	0.14 0.26	−0.08 0.54	0.05 0.71	**−0.27** **0.03**	−0.24 0.06	−0.15 0.24
*Unclass.* *Barnesiellaceae*	r p	−0.015 0.91	−0.007 0.96	−0.06 0.64	0.04 0.77	−0.015 0.91	0.14 0.29	−0.12 0.34	**0.3** **0.016**	**−0.34** **0.0066**	0.17 0.19
*Unclass. Coriobacteriaceae*	r p	0.16 0.22	**0.29** **0.02**	0.11 0.37	0.013 0.92	0.16 0.20	−0.15 0.23	−0.09 0.49	−0.067 0.60	−0.15 0.24	−0.053 0.68
*Unclass.* *Christensenellaceae*	r p	0.08 0.54	**0.28** **0.027**	−0.13 0.29	0.027 0.83	0.037 0.77	0.025 0.84	−0.21 0.11	0.0052 0.97	−0.091 0.47	−0.052 0.69
*Unclass. Mogibacteriaceae*	r p	0.11 0.41	**0.26** **0.04**	0.18 0.16	−0.17 0.18	0.053 0.68	−0.081 0.52	−0.12 0.37	0.033 0.80	−0.071 0.58	0.065 0.61
*Unclass.* *Rikenellaceae*	r p	0.061 0.64	0.1 0.44	−0.15 0.23	−0.1 0.43	−0.07 0.59	0.095 0.45	−0.15 0.25	**0.27** **0.03**	−0.2 0.11	0.18 0.15

PCs, p-cresyl sulfate; IS, indoxyl sulfate; IL, interleukin; TNF-α, tumor necrosis factor alpha; MCP-1, monocyte chemo-attractant protein-1; MDA, malondialdehyde. Significant correlations are highlighted in bold.

**Table 3 toxins-13-00472-t003:** Association of gMB species with uremic toxins, oxidative stress and inflammatory cytokines.

SPECIES		Uremic Toxins		Anti-Inflammatory Cytokines			Pro-Inflammatory Cytokines				Oxidative Stress
		IS µmol/L	PCs µmol/L	IL-10 pg/mL	IL-12 pg/mL	Fetuin-A ng/mL	IL-6 pg/mL	IL-17 pg/mL	TNF-α pg/mL	MCP-1 pg/mL	MDA µmol/L
*Lactobacillus Unclass.Lactobacillus*	r p	**0.42** **0.0005**	−0.04 0.76	0.21 0.10	0.098 0.45	0.2 0.12	−0.11 0.39	−0.02 0.88	**−0.3** **0.016**	0.09 0.48	−0.042 0.74
*Bifidobacterium-longum*	r p	**0.25** **0.047**	0.012 0.93	0.11 0.39	−0.19 0.15	0.16 0.21	−0.12 0.35	0.0056 0.97	0.11 0.40	−0.12 0.35	0.092 0.47
*Eubacterium-biforme*	r p	0.061 0.64	**0.35** **0.006**	0.12 0.33	−0.043 0.74	0.13 0.32	−0.016 0.89	−0.24 0.06	−0.019 0.88	0.11 0.39	−0.089 0.49
*Oscillospira*	r p	0.18 0.16	**0.28** **0.029**	0.083 0.51	−0.096 0.46	0.017 0.90	0.079 0.54	−0.034 0.80	0.087 0.49	−0.14 0.28	0.045 0.72
*Bacteroides-Unclass.* *Bacteroides*	r p	0.056 0.66	−0.1 0.43	**−0.26** **0.04**	−0.019 0.88	0.00004 0.99	**0.29** **0.02**	−0.15 0.26	**0.28** **0.026**	**−0.26** **0.042**	0.093 0.47
*Collinsella-aerofaciens*	r p	0.12 0.37	0.092 0.48	0.15 0.23	0.01 0.94	**0.28** **0.02**	−0.12 0.36	**0.28** **0.028**	−0.14 0.27	0.07 0.58	0.052 0.69
*Clostridium-difficile*	r p	0.13 0.30	0.059 0.65	0.14 0.27	−0.095 0.46	0.2 0.12	−0.024 0.85	−0.099 0.45	−0.058 0.65	0.054 0.67	−0.16 0.22
*Unclass. Christensenellaceae*	r p	0.074 0.57	0.29 0.021	−0.15 0.24	0.034 0.79	0.037 0.77	0.03 0.82	−0.2 0.13	−0.001 0.99	−0.064 0.61	−0.052 0.68
*Unclass. Clostridiaceae*	r p	0.21 0.098	**0.34** **0.007**	−0.01 0.94	−0.04 0.76	0.15 0.22	0.064 0.62	−0.22 0.096	0.045 0.73	−0.018 0.88	−0.09 0.46
*Unclass. Mogibacteriaceae*	rp	0.1 0.44	**0.29** **0.023**	0.15 0.22	−0.16 0.21	0.054 0.67	−0.072 0.57	−0.11 0.40	0.023 0.86	−0.038 0.76	0.063 0.62

PCs, p-cresyl sulfate; IS, indoxyl sulfate; IL, interleukin; TNF-α, tumor necrosis factor alpha; MCP-1, monocyte chemoattractant protein-1; MDA, malondialdehyde. Significant correlations are highlighted in bold.

**Table 4 toxins-13-00472-t004:** Comparison of uremic toxins, MDA and cytokines between sarcopenic and not-sarcopenic individuals.

Variables	Sarcopenic (*n* = 18)	Not Sarcopenic (*n* = 45)	*p*
*Uremic toxins*			
PCs (µmol/L)	0.65 ± 0	0.79 ± 0.18	0.24
IS (µmol/L)	1.83 ± 2.61	2.2 ± 3.13	0.43

*Pro-Inflammatory cytokines*			
IL-6 (pg/mL)	2.4 ± 4.93	2.88 ± 4.83	0.70
TNFα (pg/mL)	14.63 ± 8.84	14.14 ± 8.65	0.99
MCP-1 (pg/mL)	428.88 ± 316.49	447.17 ± 199.71	0.36
IL17 (pg/mL)	2.4 ± 4.93	2.88 ± 4.83	0.70

*Anti-Inflammatory cytokines*			
IL12p70 (pg/mL)	0.605 ± 1.80	1.471 ± 2.13	0.45
IL-10 (pg/mL)	1.08 ± 1.66	2.4 ± 5.87	0.039
Fetuin-A (ng/mL)	0.56 ± 0.65	0.64 ± 0.56	0.56

*Oxidative Stress*			
MDA (µmol/L)	0.275 ± 0.31	0.346 ± 0.18	0.47

PCs, p-cresyl sulfate; IS, indoxyl sulfate; IL, interleukin; TNFα, tumor necrosis factor alpha; MCP-1, Monocyte Chemoattractant Protein-1; MDA, malondialdehyde. Significant correlations are highlighted in bold.

## Data Availability

The data presented in this study are openly available in Dryad and FlowRepository at doi:10.1371/journal.pone.0228530 [16].

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
