# Peer review of "Association of Sarcopenia and Gut Microbiota Composition in Older Patients with Advanced Chronic Kidney Disease, Investigation of the Interactions with Uremic Toxins, Inflammation and Oxidative Stress"

_toxins, 2021, doi:10.3390/toxins13070472_

Round 1

Reviewer 1 Report

This study found the gut microbiota was significantly different, uremic toxins, inflammation and oxidative stress were not different in sarcopenic and non-sarcopenic CKD patients. This study is important as sarcopenia is prevalent in CKD patients and is associated with increased morbidity and mortality.

There are many clinical factors, as the authors listed in table 1, may affect the gut microbiota composition. We can see age with p=0.05 and BMI with p=0.025 have effects on S and NS, and I think they also may have effects on gut microbiota. So we can not distinguish the gMB difference from the S/NS or from the clinical factors, like age, BMI. In my opinion, the statistical analysis for sarcopenia and gMB composition need to perform associated clinical factors adjustment.

Author Response

Dear Reviewer,

We thank you for the in-depth revision and for your useful comments that contributed to ameliorate the overall quality of our paper. We answered below to your requests as well as we amended the manuscript where necessary.

Best regards.

Authors

This study found the gut microbiota was significantly different, uremic toxins, inflammation and oxidative stress were not different in sarcopenic and non-sarcopenic CKD patients. This study is important as sarcopenia is prevalent in CKD patients and is associated with increased morbidity and mortality.

There are many clinical factors, as the authors listed in table 1, may affect the gut microbiota composition. We can see age with p=0.05 and BMI with p=0.025 have effects on S and NS, and I think they also may have effects on gut microbiota. So we can not distinguish the gMB difference from the S/NS or from the clinical factors, like age, BMI. In my opinion, the statistical analysis for sarcopenia and gMB composition need to perform associated clinical factors adjustment.

We agree that AGE and BMI may be associated with some changes in microbiota independently of sarcopenia. However, given the cross-sectional design of the study it would be impossible to clarify the reciprocal relationships of these associations as well as to establish which ones developed first or could be considered as causal ones.

In this cohort BMI is significantly lower in sarcopenic individuals and this may depend on the correlation between sarcopenia and malnutrition that is quite a common characteristic in older patients with advanced CKD, as it was described also by our group (Sarcopenia is Associated with Malnutrition but Not with Systemic Inflammation in Older Persons with Advanced CKD. Vettoretti S, et al Nutrients. 2019 Jun 19;11(6):1378. doi: 10.3390/nu11061378).GMB composition may differ in subjects over or underweight; in order to explore this correlation, we analyzed whether there were specific associations between gMB composition and BMI (see tables 1s and 2s in supplemental materials). We found that BMI correlated negatively with Akkermansia and positively with Faecalibacterium and Paraprevotella genera. We described the correlations between BMI and Akkermansia in the text. Faecalibacterium is more abundant in obese people (PMID: 27577947 DOI: 10.2174/1871530316666160831093813; DOI: https://doi.org/10.1017/S0007114509992182) while Paraprevotella has been found in higher concentrations in the gut microbiota of sedentary young people and obese adults with NAFLD (doi.org/10.1371/journal.pone.0171352;  doi: 10.1371/journal.pone.0149564, PMID: 26919743). However, unlike Akkermansia that is more abundant in S patients and it is inversely correlated with BMI, Faecalibacterium and Paraprevotella did not differ between N and NS, therefore the correlations found between these latter two genera and BMI do not seem to be associated with the presence of sarcopenia.

We looked also for possible correlations between age and gMB composition (Table 1s and 2s) but we did not find any statistically significant correlation with bacterial families or genera.

Reviewer 2 Report

Title: Association of sarcopenia and gut microbiota composition in patients with chronic kidney disease, investigation of the interactions with uremic toxins, inflammation and oxidative stress.

In patients with CKD, loss of muscle mass or sarcopenia is affected multiple mechanisms. The current study aims to explore the association between gMB composition, uremic toxins, systemic inflammation and sarcopenia in elderly patients with CKD. Overall, the study found several correlations between gMB composition, uremic toxins, inflammatory cytokines and MDA. However, uremic toxins, inflammatory cytokines and MDA were not different in S and NS individuals, except for IL-10 that was higher in NS. The overall study can be potentially interesting especially since it may provide data on the alterations on the gMB alteration. Considering the multiplicity of abnormalities likely involved and possible, I believe that several similar studies will serve to define the gMB alteration and abnormality spectrum.

  • The main concern with the study is the possible lack of or unclear hypothesis. The authors have conducted a gMB study in a cross section of CKD population and are reporting the positive findings.
  • I believe that if this is a descriptive paper, it is uncertain that it provides the full context. The investigators have provided correlative statistics for a number of microbial species based on what is considered significant effectors but, it is uncertain as to what is the total number of species examined. I think a full microbial spectrum and microbiomical representation needs to be provided to ensure that we examine the true significance of the associations found in the handful of bacteria.
  • Additionally, even among the positive findings, the strength of association appears to be marginal, especially considering the adjustments for the multiple outcome analysis.
  • Finally, these multiple analyses thus lead to a large discussion about the individual bacterium/s and the adverse inflammatory or sarcopenic phenotype. However, considering the multiplicity of the analysis and relative independent discussions, it appears that a full hypothesis based is not feasible. I suggest that the authors synthesize the findings into some common messages. Considering that with the small no. of patients if this is not feasible, it may be more beneficial for this paper to be a more descriptive paper rather than focusing on a proven vs not proven hypothesis.
  • At the least this requires, a significant discussion on the limitation of the study highlighting these aspects.
  • Minor concerns:
    • Abstract: short forms for IS, PC, and all other inflammatory need full forms, if not possible, may consider batching them together, atleast for the IS, PC EWGSOP2, the uncommon ones.
    • Patient characteristics needs to have bicarbonate concentration.
    • Not sure why these two statements are separate in the abstract: “S patients had a greater abundance of Micrococcaceaee and 12 Verrucomicrobiaceaee; they had also a greater abundance of Megasphaera, Rothia, Veillonella, Ak- 13 kermansia and Coprobacillus genera.: They all say the greater abundance of several microbiota.
    • S & NS patients appear not easy to follow short form. Not sure they need it. Sarcopenia is one word and should not affect the count, if that is what they are worried for.
    • Page 1 para 2: ‘Specifically’ is not needed. Sentences are too large and each mechanism reported seprarately will read better.
    • What is scheme 18 in table 4? Is this sarcopenia group?

Author Response

Dear Reviewer,

We thank you for the in-depth revision and for your useful comments that contributed to ameliorate the overall quality of our paper. We answered below to your requests as well as we amended the manuscript where necessary.

Best regards.

Authors

Title: Association of sarcopenia and gut microbiota composition in patients with chronic kidney disease, investigation of the interactions with uremic toxins, inflammation and oxidative stress.

In patients with CKD, loss of muscle mass or sarcopenia is affected multiple mechanisms. The current study aims to explore the association between gMB composition, uremic toxins, systemic inflammation and sarcopenia in elderly patients with CKD. Overall, the study found several correlations between gMB composition, uremic toxins, inflammatory cytokines and MDA. However, uremic toxins, inflammatory cytokines and MDA were not different in S and NS individuals, except for IL-10 that was higher in NS. The overall study can be potentially interesting especially since it may provide data on the alterations on the gMB alteration. Considering the multiplicity of abnormalities likely involved and possible, I believe that several similar studies will serve to define the gMB alteration and abnormality spectrum.

  • The main concern with the study is the possible lack of or unclear hypothesis. The authors have conducted a gMB study in a cross section of CKD population and are reporting the positive findings.

I believe that if this is a descriptive paper, it is uncertain that it provides the full context. The investigators have provided correlative statistics for a number of microbial species based on what is considered significant effectors but, it is uncertain as to what is the total number of species examined. I think a full microbial spectrum and microbiomical representation needs to be provided to ensure that we examine the true significance of the associations found in the handful of bacteria.

We thank the reviewer for this comment that gave us the opportunity to improve the presentation of our results. This study was principally aimed to explore whether there was any correlation between gMB composition and sarcopenia in older patients with advanced CKD.

As secondary endpoints, we investigated whether there was any correlation of gMB composition with: inflammation, uremic toxins and oxidative stress. Furthermore, we evaluated if any of the latter variables were associated with sarcopenia.

In order to explore that we adopted quite restrictive selection criteria to exclude possible sources of bias. 

We evaluated gMB composition of our patients by examining a total of 8 phyla, 34 families and 52 genera. Those associations are summarized in Figures 1 and 2 while full analyses are reported in Figure1-2s and in Tables 1- 4s of the supplemental materials. All data refers to direct correlations without any adjustment for other covariates.

These aspects have been added to the result section (lines 104-107).

  • Additionally, even among the positive findings, the strength of association appears to be marginal, especially considering the adjustments for the multiple outcome analysis.

  • Finally, these multiple analyses thus lead to a large discussion about the individual bacterium/s and the adverse inflammatory or sarcopenic phenotype. However, considering the multiplicity of the analysis and relative independent discussions, it appears that a full hypothesis based is not feasible. I suggest that the authors synthesize the findings into some common messages. Considering that with the small no. of patients if this is not feasible, it may be more beneficial for this paper to be a more descriptive paper rather than focusing on a proven vs not proven hypothesis.
  • At the least this requires, a significant discussion on the limitation of the study highlighting these aspects.

We agree with the reviewer that this is an observational study and that it cannot provide more than the description of the associations that were found. In order to remark this we added several sentences in the discussion (lines 356-371)

  • Minor concerns:
    • Abstract: short forms for IS, PC, and all other inflammatory need full forms, if not possible, may consider batching them together, at least for the IS, PC EWGSOP2, the uncommon ones.

We amended abstract’s text accordingly

    • Patient characteristics needs to have bicarbonate concentration.

We added bicarbonate concentration in table 1.

    • Not sure why these two statements are separate in the abstract: “S patients had a greater abundance of Micrococcaceaee and 12 Verrucomicrobiaceaee; they had also a greater abundance of Megasphaera, Rothia, Veillonella, Ak- 13 kermansia and Coprobacillus genera.: They all say the greater abundance of several microbiota.

We wrote two separate sentences because the first refers to bacterial families and the second to genera. We corrected the text by adding the word "families".

    • S & NS patients appear not easy to follow short form. Not sure they need it. Sarcopenia is one word and should not affect the count, if that is what they are worried for.

We amended the text with the extended form.

    • Page 1 para 2: ‘Specifically’ is not needed. Sentences are too large and each mechanism reported separately will read better.

We removed “Specifically” and shortened sentences as suggested.

    • What is scheme 18 in table 4? Is this sarcopenia group?

Yes it refers to sarcopenic patients, we ameliorated table 4